# *Bacillus subtilis* PS-216 Antagonistic Activities against *Campylobacter jejuni* NCTC 11168 Are Modulated by Temperature, Oxygen, and Growth Medium

**DOI:** 10.3390/microorganisms10020289

**Published:** 2022-01-26

**Authors:** Katarina Šimunović, Polonca Stefanic, Anja Klančnik, Andi Erega, Ines Mandic Mulec, Sonja Smole Možina

**Affiliations:** 1Chair of Biotechnology, Microbiology, and Food Safety, Department of Food Science and Technology, Biotechnical Faculty, University of Ljubljana, 1000 Ljubljana, Slovenia; katarina.simunovic@bf.uni-lj.si (K.Š.); anja.klancnik@bf.uni-lj.si (A.K.); andi.erega@bf.uni-lj.si (A.E.); 2Chair of Microbial Ecology and Physiology, Department of Microbiology, Biotechnical Faculty, University of Ljubljana, 1000 Ljubljana, Slovenia; Polonca.Stefanic@bf.uni-lj.si (P.S.); ines.mandicmulec@bf.uni-lj.si (I.M.M.)

**Keywords:** *Campylobacter jejuni*, *Bacillus subtilis* PS-216, bacterial interaction, *C. jejuni* reduction, chicken intestinal content, chicken litter medium

## Abstract

As the incidence of *Campylobacter jejuni* and campylobacteriosis grows, so does the need for a better understanding and control of this pathogen. We studied the interactions of *C. jejuni* NCTC 11168 and a potential probiotic, *Bacillus subtilis* PS-216, in cocultures at different starting ratios and temperatures (20 °C, 37 °C, 42 °C), under different atmospheres (aerobic, microaerobic), and in different growth media (Mueller–Hinton, chicken litter medium, chicken intestinal-content medium). Under microaerobic conditions, *B. subtilis* effectively inhibited the growth of *C. jejuni* at 42 °C (log reduction, 4.19), even when *C. jejuni* counts surpassed *B. subtilis* by 1000-fold in the starting inoculum. This inhibition was weaker at 37 °C (log reduction, 1.63), while no impact on CFUs was noted at 20 °C, which is a temperature nonpermissive of *C. jejuni* growth. Under aerobic conditions, *B. subtilis* supported *C. jejuni* survival. *B. subtilis* PS-216 inhibited the growth of *C. jejuni* in sterile chicken litter (4.07 log reduction) and in sterile intestinal content (2.26 log reduction). In nonsterile intestinal content, *B. subtilis* PS-216 was able to grow, to a lesser extent, compared to Mueller–Hinton media, still showing potential as a chicken probiotic that could be integrated into the chicken intestinal microbiota. This study showed the strong influence of environmental parameters on the variability of *C. jejuni* and *B. subtilis* interactions. Furthermore, *B. subtilis* PS-216 antagonism was strongest against *C. jejuni* NCTC 11168 under conditions that might represent conditions in the chicken environment (42 °C, microaerobic atmosphere, chicken litter medium).

## 1. Introduction

*Campylobacter jejuni* is a major cause of the most commonly reported bacterial gastroenteritis, campylobacteriosis, and it is considered a serious food safety hazard [1]. Campylobacteriosis manifests as acute watery diarrhea, fever, and cramps [2,3,4]. *C. jejuni* is a burden on both health and the economy, as the cost of campylobacteriosis and its consequences, estimated by EFSA on the basis of disability-adjusted life years per year, reaches EUR2.4 billion/year [1,5]. There is, thus, the need for better and more effective control measures.

Infection with *C. jejuni* is highly associated with undercooked poultry, as this pathogen is a commensal in the poultry gut [1,2]. Indeed, reduction of *Campylobacter* spp. by 3 log_10_ units in the chicken cecum at slaughter can reduce the public health risk by up to 90%, and therefore, great efforts are being made to reduce *Campylobacter* spp. in poultry production [5,6,7].

As part of the avian gut microbiota, *C. jejuni* is exposed to interactions with other inhabitants of the gut, which can either compete for nutrients and place, or cooperate, and thus ensure mutual benefit. Competition, rather than cooperation, is more common amongst different bacterial species [8], and the evidence suggests that when a nonpathogenic strain is introduced into the niche of a pathogenic *C. jejuni* strain, it might outcompete it, and thus remove it from that niche [9]. This concept is known as competitive exclusion, and it is often exploited using novel probiotic species or strains with the potential to control pathogens, such as *C. jejuni* [10,11]. As the primary niche of *C. jejuni* is the avian gut, efforts are focused on the reduction of this pathogen in this important source.

Probiotics have been shown to aid in the reduction of *C. jejuni* in broiler chickens, and bacteria isolated from the chicken gut are often investigated for this purpose, including *Lactobacillus* and *Bacillus* spp. [12,13,14,15]. Different *Bacillus subtilis* strains have been studied for the reduction of *C. jejuni* in poultry production, with promising results, and with this effect being strain specific, as not all *B. subtilis* can reduce *C. jejuni* levels [13]. *B. subtilis* strains as feed supplements can modify the chicken gut microbiota, and the average chicken body weight and gut health [15,16]. *B. subtilis* has the potential to produce different antibacterial substances, which might be one of the mechanisms they use against pathogenic bacteria [17,18,19].

Bacterial interactions are complex, even in a two-species system [20], and environmental parameters affect these interactions, such as temperature and oxygen. In a previous study, we reported that *B. subtilis* PS-216 shows antagonism towards *C. jejuni* in the form of growth inhibition and biofilm reduction, and that this is mediated by the antimicrobial compound bacillaene [21]. Thus, we saw it as pertinent to consider their interactions under different environmental conditions, to better understand the span of activity of this potential probiotic.

Oxygen is a key parameter that can limit *C. jejuni* growth and survival in any environment, as *C. jejuni* requires 2% to 10% oxygen for growth, although it is sensitive to higher oxygen concentrations [22]. Although higher oxygen environments within the farm and slaughter house environments are toxic for *C. jejuni* survival, this pathogen can still persist as part of a consortium where it is protected by other species that deplete the oxygen in the environment, thus supporting the spread of *C. jejuni* [23,24,25]. On the other hand, although *B. subtilis* has been considered to be a strict aerobe in the past, native *B. subtilis* strains can grow under oxygen limitation by using nitrate or nitrite as a terminal electron acceptor, or by fermentation (for reviews, see [26,27]). Thus, we hypothesized that the interactions of *C. jejuni* and *B. subtilis* might be relevant in an environment with oxygen limitation, such as the animal gut, and in oxygenated environments (e.g., feathers and other surfaces of the animals colonized with *C. jejuni,* in the farm water supply, on surfaces in the slaughter house surfaces, and other environments) where interactions with *Bacillus* strains might even be beneficial for this pathogen.

Temperature is another key ecological factor that modulates bacterial growth. *C. jejuni* grows best at 42 °C, although metabolic activity in the form of ATP generation, protein synthesis, and oxygen consumption can be detected at temperatures as low as 4 °C; however, *C. jejuni* shows no active growth at <32 °C [28]. The growth of *B. subtilis* can be detected from 4 °C to 55 °C, although the optimal temperature is dependent on the strain [29,30]. Moreover, temperature can affect the production of antimicrobial compounds [31,32], which might have consequences for antagonism of *B. subtilis* against other species, including pathogens. *C. jejuni* is a fastidious microorganism that shows many limitations to its growth, including growth in a microaerobic atmosphere (i.e., limited oxygen), at limited growth temperatures (32–45 °C), and its inability to use glucose and other similar sugars as a carbon source [22,28]. *B. subtilis* is more flexible in terms of its growth requirements, although it is not known whether *B. subtilis* antagonizes *C. jejuni* under conditions that do not support *C. jejuni* growth (e.g., 20 °C). Although the interactions of these two species are likely to occur in many environments, the information on how changed environmental conditions will affect these interactions is limited.

To evaluate *B. subtilis* PS-216 as a probiotic against *C. jejuni* in the chicken host, we first explored and defined their cocultivation dynamics at 42 °C, 37 °C, and 20 °C, under microaerobic versus aerobic conditions, and in the more complex media of chicken litter and chicken intestinal content, which represent environments that are closer to the conditions in the chicken gut. We show that *B. subtilis* acts antagonistically against *C. jejuni* at the higher temperatures, under microaerobic conditions, and in the complex media, but is protective under otherwise unfavorable conditions for *C. jejuni* at reduced temperatures and under aerobic conditions. These data also suggest that the complex media introduced here represent a model for initial evaluation of bacterial interactions in the chicken gut.

## 2. Materials and Methods

### 2.1. Strains and Growth Conditions

All cultures were stored at −80 °C in Mueller–Hinton broth (MHB; Oxoid, UK) with 20% glycerol. From frozen stock solutions, *C. jejuni* NCTC 11168 was incubated on Karmali agar (Oxoid, UK) supplemented with Karmali selective supplement (SR0502; Oxoid, UK) for 24 h at 42 °C under microaerobic conditions (5% O_2_, 10% CO_2_, 85% N_2_). The microaerobic atmosphere was obtained by flushing the containers for the anaerobic incubation of samples (anaerojar) with the appropriate gas mixture just before the start of the incubations. *B. subtilis* PS-216 was originally isolated from riverbank soil of the Sava River in Slovenia [33], and was grown here on Mueller–Hinton agar (MHA; Oxoid, UK) at 37 °C under aerobic conditions for 24 h.

For cocultivation experiments, *C. jejuni* NCTC 11168 was further transferred onto MHA (second subculture) and incubated at 42 °C under microaerobic conditions for 24 h. *B. subtilis* PS-216 was further transferred onto MHA and incubated at 37 °C under aerobic conditions (second subculture) for 24 h. From the second subculture of the *C. jejuni* NCTC 11168 and *B. subtilis* PS-216, cultures with OD_600_ 0.1 (approx. 5 × 10^7^ CFU/mL and 5 × 10^6^ CFU/mL, respectively) were prepared in MHB and used as the inoculum in all of the experiments.

*Campylobacter**jejuni* was enumerated on Karmali agar supplemented with Karmali selective supplement after 24 h of incubation at 42 °C under microaerobic conditions. *B. subtilis* was enumerated on MHA after 24 h of incubation at 37 °C under aerobic conditions. *Bacillus* spp. from fecal samples were enumerated on HiChrom *Bacillus* agar (Himedia, USA). To enumerate the *B. subtilis* spores, a protocol by Štefanič and Mandič-Mulec (2009) [33] was used, with some modifications. Specifically, 1.0 mL samples were transferred from the experimental cultures into epitubes and incubated at 80 °C for 30 min in a water bath, then left to cool down at room temperature. The samples were then serially diluted in phosphate-buffered saline, and plated onto HiChrom *Bacillus* agar. The number of vegetative cells in the samples was determined by subtracting the number of spores from the untreated samples.

### 2.2. Cocultivation of B. subtilis with C. jejuni in Mueller–Hinton Broth at Different Starting Ratios

*Bacillus subtilis* PS-216 was cultivated with *C. jejuni* NCTC 11168 at starting ratios (*B. subtilis*:*C. jejuni*) of 10:1 (approx. 10^4^ CFU/mL:10^3^ CFU/mL), 1:10 (approx. 10^2^ CFU/mL:10^3^ CFU/mL), 1:100 (approx. 10^4^ CFU/mL:10^6^ CFU/mL), and 1:10,000 (approx. 10^2^ CFU/mL:10^6^ CFU/mL) at 42 °C under microaerobic conditions in MHB (Oxoid, UK) for 48 h. The cultures were prepared from OD_600_ 0.1 inocula and the numbers of each of the strains were determined as CFU/mL at the start (t = 0 h), and at the sampling times of 4, 8, 12, 24, and 48 h. As controls, *B. subtilis* and *C. jejuni* monocultures were prepared at the same starting concentrations as in the cocultures. Three independent experiments were carried out, with up to three technical replicates, and the data are shown as means ± standard deviation.

### 2.3. Cocultivation of B. subtilis with C. jejuni in Mueller–Hinton Broth at Different Temperatures

The effects of different temperatures on the growth dynamics of *B. subtilis* PS-216 and *C. jejuni* NCTC 11168 in cocultures were tested at 42 °C, 37 °C, and 20 °C under microaerobic conditions for 48 h. The cultures were prepared at 1:10 starting ratios in MHB, as approx. 10^4^ CFU/mL:10^5^ CFU/mL for 42 °C and 37 °C, and approx. 10^5^ CFU/mL:10^6^ CFU/mL for 20 °C (*B. subtilis*:*C. jejuni*). The concentrations of the bacterial cells were determined at the beginning (t = 0 h), and after 4, 24, and 48 h of incubation under microaerobic conditions, by disturbed sampling. Three independent experiments were carried out, with up to three technical replicates, and the data are shown as means ± standard deviation.

### 2.4. Cocultivation of B. subtilis with C. jejuni in Mueller–Hinton Broth under Aerobic Conditions

The effects of oxygen on the dynamics of the cocultures were tested by cultivation of *B. subtilis* PS-216 and *C. jejuni* NCTC 11168 under aerobic conditions at 37 °C and 20 °C for 48 h. The cultures were prepared at 1:10 starting ratios in MHB (*B. subtilis*:*C. jejuni;* approx. 10^4^ CFU/mL:10^5^ CFU/mL for 37 °C, and approx. 10^5^ CFU/mL:10^6^ CFU/mL for 20 °C). The concentrations of the bacterial cells were determined at the beginning (t = 0 h), and after 4, 24, and 48 h of incubation, by disturbed sampling. Three independent experiments were carried out, with up to three technical replicates, and the data are shown as means ± standard deviation.

### 2.5. Cocultivation of B. subtilis PS-216 and C. jejuni NCTC 11168 in Broiler Chicken Intestinal Content Medium

*Bacillus subtilis* PS-216 and *C. jejuni* NCTC 11168 were cultivated in two sterile broiler chicken feces media models: (i) a sterile litter medium; and (ii) a sterile intestinal-content medium. The starting ratio was 1:100 (*B. subtilis*:*C. jejuni;* approx. 10^3^ CFU/mL:10^5^ CFU/mL), with incubation at 42 °C under microaerobic conditions for 48 h.

The broiler chicken litter medium was prepared from broiler chicken litter collected from a commercial farm (Perutnina Ptuj d.d., Slovenia), which was suspended in phosphate-buffered saline (Oxoid, UK) at a ratio of 1:9 (*w/v*), and then autoclaved (121 °C, 15 min). For the broiler chicken intestinal-content medium, the intestinal contents (small and large intestines combined) from 14 broiler chickens from one flock of 49-day-old broilers from a semi-intensive rearing system (fed with bro-starter and bro-finisher obtained from Jata Emona d.o.o., Slovenia), were collected, pooled, and stored at −80 °C. For these cultivation experiments, the intestinal content was diluted in phosphate-buffered saline at a ratio of 1:1 (*v/v*) and autoclaved. *B. subtilis* and *C. jejuni* monoculture positive controls were used, along with a negative control of the prepared medium without the bacteria added. The concentrations of bacterial cells were determined at the beginning (t = 0 h), and after 8, 24, and 48 h of incubation at 42 °C under microaerobic conditions, by disturbed sampling, where tubes containing the samples were taken from the microaerobic atmosphere, mixed, sampled, and then returned to the incubations under the appropriate conditions. All of the samples throughout all of the incubations were taken from the same tube. Three biological replicates were carried out, with up to three technical replicates.

### 2.6. Growth of B. subtilis PS-216 in Nonsterile Broiler Chicken Intestinal Content Medium

To determine the suitability of *B. subtilis* PS-216 for use in broiler chickens, its growth was determined in a nonsterile broiler chicken intestinal-content medium. For this purpose, the broiler chicken intestinal content was diluted in phosphate-buffered saline at a ratio of 1:1 (*v/v*), and homogenized (but not sterilized). The initial *B. subtilis* PS-216 inoculum was approx. 5 × 10^2^ CFU/mL, which consisted of 2 × 10^2^ CFU/mL vegetative cells and 3 × 10^2^ CFU/mL spores. As a background control, the medium was also prepared without added *B. subtilis*. The concentrations of the vegetative bacterial cells and spores were determined at the beginning (t = 0 h), and after 8, 24, and 48 h of incubation under microaerobic conditions in the inoculated samples and in the control samples without inoculated *B. subtilis*. Three biological replicates were carried out, with up to three technical replicates. The vegetative cell counts were obtained by subtracting the numbers of spores (after heat treatment at 80 °C for 15 min) determined from the total number of colonies from the untreated samples, and the background growth (uninoculated samples) was considered in the calculations.

### 2.7. Statistical Analysis

To evaluate the influence of cocultivation on *B. subtilis* and *C. jejuni* growth, Mann–Whitney U-tests were used. All of the analyses were performed with SPSS software, version 22 (IBM Corp., Armonk, NY, USA) and *p* < 0.05 was considered as statistically significant.

## 3. Results

### 3.1. B. subtilis PS-216 Reduces C. jejuni Growth

The interactions of *C. jejuni* NCTC 11168 and *B. subtilis* PS-216 were determined during cocultivation.

We first cocultured *B. subtilis* and *C. jejuni* in MHB at the *B. subtilis*:*C. jejuni* ratios of 10:1 and 1:10. After 4 h and 8 h, *B. subtilis* did not show any detectable effects on *C. jejuni* growth, regardless of the *B. subtilis*:*C. jejuni* ratio (Figure 1A,B). However, after 24 h at both the 10:1 and 1:10 ratios, the presence of *B. subtilis* resulted in reduced *C. jejuni* counts (log reduction, 3.87, 4.07, respectively). After 48 h, the *C. jejuni* counts were below the limit of detection for both of these cocultures (Figure 1A,B).

When *B. subtilis* was outnumbered by *C. jejuni* by 100-fold in the starting inoculum (i.e., *B. subtilis*:*C. jejuni*, 1:100), the *C. jejuni* counts were reduced by 0.91 log units after 24 h, and by 4.22 log units after 48 h (Figure 1C). The anti-*Campylobacter* effects of *B. subtilis* were weaker when *B. subtilis* was initially 10,000-fold lower (*B. subtilis*:*C. jejuni*, 1:10,000), as no significant reduction in *C. jejuni* was seen after 24 h (Figure 1D, log reduction, 0.32). However, after 48 h, the *C. jejuni* counts here were significantly decreased by 1.62 log units. This shows that *B. subtilis* is strongly competitive, as the growth inhibition of *C. jejuni* was seen even when *B. subtilis* was greatly outnumbered by *C. jejuni*. On the other hand, the growth of *B. subtilis* was not affected by the presence of *C. jejuni*, regardless of the initial inoculum ratios (Appendix A).

### 3.2. B. subtilis PS-216 Anti-Campylobacter Effects Increase at Higher Temperatures

We investigated the interactions of *B. subtilis* PS-216 and *C. jejuni* NCTC 11168 in MHB at the starting ratio of 1:10 (*B. subtilis*:*C. jejuni*) and under microaerobic conditions, at 42 °C (to mimic the chicken gut), 37 °C (to mimic the human gut), and 20 °C (as the room/environment temperature). Here, the presence of *B. subtilis* PS-216 significantly reduced *C. jejuni* growth at 42 °C and 37 °C, while no impact on C. jejuni NCTC 11168 survival was noted at 20 °C (Figure 2A–C). After 24 h at 42 °C, the *C. jejuni* counts in the co-cultures were reduced by 4.19 log units (*p* < 0.05), compared to the monoculture, and after 48 h by >6.5 log units (Figure 2A). At 37 °C, the *C. jejuni* counts after 24 h and 48 h were reduced by 1.63 log units and 2.68 log units, respectively (*p* < 0.05, for both), as compared to the control monoculture (Figure 2B). The effects of *B. subtilis* were weaker at 37 °C compared to 42 °C under these microaerobic conditions, but remained significant. In contrast, at 20 °C, *B. subtilis* PS-216 did not show any significant anti-*Campylobacter* effects (Figure 2C). These data show that the environmental temperature has a crucial role in the interactions between *B. subtilis* and *C. jejuni*. For the *B. subtilis* PS-216 counts after 48 h, although they were a little higher in the monoculture, they again showed no significant differences between the mono- and cocultures. They were, however, significantly higher at 42 °C and 37 °C, compared to 20 °C (Figure 2D).

### 3.3. Protective Effects of B. subtilis PS-216 for C. jejuni under Aerobic Conditions

Here, we tested the effects of *B. subtilis* PS-216 on the survival of *C. jejuni* NCTC 11168 at the starting ratio of 1:10 (*B. subtilis*:*C. jejuni*) at 37 °C and 20 °C under aerobic growth conditions, representing conditions found on the surface and surroundings of broiler chickens. Protective, rather than inhibitory, effects of *B. subtilis* PS-216 on *C. jejuni* were seen in these cocultures at both 37 °C and 20 °C (Figure 3). Under these aerobic conditions, *C. jejuni* did not undergo growth in the mono- or cocultures; indeed, after 48 h in the monoculture at 37 °C, no *C. jejuni* was detected, whereas the *C. jejuni* counts in the coculture remained comparable to the starting inoculum (Figure 3A). Again, no significant differences were seen for *B. subtilis* PS-216 growth in the cocultures, compared to monocultures (Figure 3).

### 3.4. Anti-Campylobacter Effects of B. subtilis PS-216 Are More Prominent in Chicken Litter Than in Intestinal Content

To evaluate the anti-*Campylobacter* effects of *B. subtilis* PS-216 in an environment closer to the conditions where these two bacteria might interact, a sterile chicken litter medium and a chicken intestinal-content medium were prepared. Chicken litter is a complex mixture of excreted chicken fecal matter and cellulose material. In the chicken litter medium, *C. jejuni* growth was strongly inhibited by *B. subtilis* (Figure 4A). After 8 h of this incubation with *B. subtilis*, the *C. jejuni* counts were significantly reduced by 1.18 log units, and after 24 h, no *C. jejuni* was detected in the chicken litter medium. This anti-*Campylobacter* effect of *B. subtilis* PS-216 was stronger in the chicken litter medium compared to the effects in MHB: in the *B. subtilis* PS-216 cocultures (starting ratio, 1:10) after 24 h, the log reductions in *C. jejuni* counts in MHB (Figure 1B) and in the sterilized chicken litter medium (Figure 4A) were 4.07 and >6, respectively. The *B. subtilis* growth in the sterilized chicken litter medium was not affected by the presence of *C. jejuni* (Appendix A), and was comparable with MHB.

In the chicken intestinal-content medium, the *B. subtilis* cocultures did not show any bactericidal effects against *C. jejuni*, as *C. jejuni* continued to grow for 48 h even in the presence of *B. subtilis* (Figure 4B), although at a significantly slower rate than in the monoculture. Here, *B. subtilis* reduced the *C. jejuni* counts by 2.26 log units at 24 h, and 1.84 log units at 48 h of coculture. The less dramatic effect of *B. subtilis* here can be explained by its hindered growth in the sterile intestinal-content medium (Appendix A).

### 3.5. B. subtilis PS-216 Growth and Sporulation Are Limited in Nonsterile Chicken Intestinal Content Medium

We cultivated *B. subtilis* PS-216 in the nonsterile chicken intestinal-content medium to follow its growth and sporulation for 48 h at 42 °C under microaerobic conditions. After an initial drop in the vegetative cell counts due to sporulation, the *B. subtilis* grew to the final count of 4.54 log_10_CFU/mL after 48 h of incubation (Figure 5). These cells did not sporulate, as after 24 h the spore counts were below, and remained below, the limit of detection.

The interaction of *C. jejuni* and *B. subtilis* was not examined under these conditions, as *C. jejuni* did not survive in the nonsterile intestinal content in the monoculture inoculated at 10^4^ CFU/mL and 10^6^ CFU/mL (data not shown).

## 4. Discussion

The persistence of *C. jejuni* in poultry production represents a considerable food safety hazard, and is, thus, an urgent problem that needs to be resolved. Although safety measures are being implemented, the need for additional interventions persists. One potential intervention is the addition of probiotic bacteria that can substantially reduce the *C. jejuni* load in the chicken host [1,5,6]. *B. subtilis* is a spore-forming bacterium that is often found in the chicken gut and is being investigated for probiotic use, with varying results. However, little is known about how environmental parameters influence the interactions between *B. subtilis* and *C. jejuni*.

Thus, we here characterized the interactions between *B. subtilis*-PS-216 and *C. jejuni* NCTC 11168 in laboratory media in terms of temperature and the atmosphere of the cultivation, and also in the more complex environments of a chicken litter medium and a chicken intestinal-content medium. All of the tested parameters influenced the active growth and survival of *C. jejuni* when in coculture with *B. subtilis*.

Specifically, at 42 °C and under the microaerobic conditions that favor *C. jejuni* growth, *B. subtilis* PS-216 inhibited *C. jejuni* growth even when *B. subtilis* was initially strongly outnumbered by *C. jejuni*. This could be important in the chicken gut, where *C. jejuni* can be found in high numbers of up to 10^9^ CFU/g feces [34,35], although additional in vivo experiments must be performed to confirm this. This growth inhibition of *C. jejuni* by *B. subtilis* PS-216 is consistent with our previous findings [21]. Here, this was extended to interaction studies also at 37 °C and 20 °C; this *B. subtilis* PS-216 growth inhibitory effect on *C. jejuni* at 42 °C was reduced already at 37 °C, and then lost at 20 °C (Figure 2). Indeed, no active growth of *C. jejuni* is expected at temperatures <32 °C, although the metabolic activity of *C. jejuni* has been reported at temperatures as low as 4 °C [28]. Our results thus indicate that for strong anti-*Campylobacter* effects of *B. subtilis* seen as a reduction of *C. jejuni* numbers as compared to the control monoculture, active growth of *C. jejuni* is required. The incubation temperature can also affect the *B. subtilis* antimicrobial synthesis and antagonistic behavior towards other species [31,32], which might also apply to these cocultivations with *C. jejuni*. Moreover, higher growth inhibition of *B. subtilis* PS-216 at higher temperatures suggests that this *B. subtilis* strain might have greater benefits in the chicken host (i.e., at 42 °C) than in a human host (i.e., at 37 °C). However, the gastrointestinal tract of animals is a complex environment and the in vitro inhibition data might not translate to in vivo settings. 

*Campylobacter jejuni* growth is strongly limited by the presence of oxygen. *C. jejuni* requires an oxygen tension of 2% to 10% for growth, and it is also sensitive to higher oxygen concentrations [22]; in contrast, *B. subtilis* can thrive under these conditions. Here, we showed that the interactions between *B. subtilis* and *C. jejuni* also changed drastically under aerobic conditions, as *B. subtilis* supported the survival of *C. jejuni* at 37 °C and 20 °C (Figure 3). Similar phenomena were described by Ica et al. [36] and Hilbert et al. [24], where they reported that other species (i.e., *Pseudomonas* spp., *Staphylococcus* spp.) can protect *C. jejuni* from harmful atmospheric conditions. Indeed, a good example of this was provided by Culotti and Packman [23], where *C. jejuni* grew in biofilms under aerobic conditions in cocultures with *Pseudomonas aeruginosa*. Here, it was indicated that the active growth and respiration of *Pseudomonas* spp. can lower the oxygen concentration of the local environment, and thus provide protection for *C. jejuni* [23]. As anti-*Campylobacter* effects will only be apparent when *C. jejuni* is actively growing, the protective effects seen here for *B. subtilis* under aerobic conditions are not surprising. Under these conditions, where oxygen levels would be toxic to *C. jejuni*, there would be no obvious antagonistic effects, although *B. subtilis* would still reduce oxygen in the environment. Therefore, this could be a food safety concern, as it could promote the survival and persistence of *C. jejuni* in the environment.

The reduction of *C. jejuni* in broiler chickens using a probiotic such as *B. subtilis* might represent a promising tool in the control of *C. jejuni*. Fritts et al. [37] reported that *B. subtilis* C-3120 (known as Calsporin) can slightly reduce *C. jejuni* on broiler carcasses (log reduction, 0.2), and later, Guyard-Nicodeme et al. [38] confirmed the reduction of *C. jejuni* in the cecum content (1.7 log reduction). This anti-*Campylobacter* effect was not seen for *B. subtilis* DSM17299 [39]. This suggests that *B. subtilis* has strain-specific anti-*Campylobacter* activity, and thus, there is the need to better characterize the interactions of these two organisms to evaluate their potential as new probiotics for *C. jejuni* control.

As cocultivation of *B. subtilis* and *C. jejuni* at 42 °C under microaerobic conditions resulted in reductions in *C. jejuni* growth, we also used simple model systems of chicken litter and chicken intestinal content to evaluate the potential *B. subtilis* anti-*Campylobacter* activity in poultry. These two media had different compositions. The litter was a mixture of chicken excreta and cellulose material from used bedding, whereas the intestinal content contained less undigested cellulose material. In the medium containing sterile chicken litter, *B. subtilis* growth was comparable to its growth in the rich laboratory medium of MHB (Appendix A), while its anti-*Campylobacter* effects were much stronger. This excellent growth of *B. subtilis* PS-216 in the sterile chicken litter medium might also be due to the heat treatment of cellulose material from the bedding and possible subsequent release of sugars [40]. The intestinal content is expected to have less cellulose material, and thus, fewer nutrients compared to the chicken litter medium after autoclaving. In the chicken intestinal-content medium, the *B. subtilis* anti-*Campylobacter* effects were not clearly bactericidal, although significant reduction of *C. jejuni* growth was still obtained, compared to the untreated control (Figure 4). The reduced effects of *B. subtilis* here would not be affected by hindered *C. jejuni* growth, as it was comparable to the growth in laboratory media, but could rather be explained by the hindered growth of *B. subtilis* in the sterile chicken intestinal-content medium (Appendix A), and thus resulting in a reduced killing effect. As the compositions of chicken bedding (and, thus, litter) and also the intestinal content of different animals can be variable, this interaction of *B. subtilis* and *C. jejuni* in the media from chicken litter and intestinal content seen in this study might be expected to vary accordingly.

*Bacillus subtilis* strains have been shown to have enzymes for the breakdown of complex materials such as cellulose, xylan, and levan, and to grow in highly competitive environments such as chicken feces [41,42,43,44,45]. The hydrolysis of indigestible complex carbohydrates to simple sugars by bacteria in the chicken gut allows for their further fermentation and use by the host as energy and carbon sources. This can further feed conversion and growth of broiler chickens [45], making research into *B. subtilis* as a probiotic interesting from the perspectives of animal health and growth enhancement, as well as antipathogenic effects.

The finding that active growth of *C. jejuni* might be required for the inhibitory effects of *B. subtilis* makes *B. subtilis* suitable for *C. jejuni* control in the gut, where *C. jejuni* will replicate. Conversely, *B. subtilis* does not appear to be suitable as a *C. jejuni* antagonist for environments rich in oxygen or at lower temperatures, as are often found on exposed surfaces such as the ground, water feeders, or feathers. Indeed, this might represent a challenge, as *B. subtilis* might even promote the survival of *C. jejuni* in aerobic environments, and thus contribute to the persistence of *C. jejuni* in the environment instead, although more studies are necessary to confirm this.

For a *B. subtilis* strain to be effective against *C. jejuni* in an environment with a naturally occurring high microbiological burden such as the chicken gut, it has to be resistant to the negative effects of this environment, and thus, be competitive and grow here. *B. subtilis* PS-216 actively grew in the nonsterile chicken intestinal-content medium, where it had to compete with the normal chicken intestine microbiota under oxygen-limiting conditions. Interestingly, the *B. subtilis* spores germinated in the chicken intestinal-content medium, although no sporulation occurred, even after 48 h of incubation (Figure 5). Cartman et al. (2008) [46] have also shown that *B. subtilis* spores can germinate in the chicken gut. They showed that at 20 h postexposure with *B. subtilis* spores, the numbers of vegetative cells were higher compared to the spore counts. These in vivo findings are consistent with the present in vitro results, and strongly suggest that *B. subtilis* spores will germinate in the chicken gut, and that cell replication will occur.

The good anti-*Campylobacter* activity shown here under conditions close to the chicken gut allow us to propose *B. subtilis* PS-216 as a promising candidate for *C. jejuni* control, and potentially as an interesting probiotic for use in poultry production.

## 5. Conclusions

*Bacillus subtilis* PS-216 has shown strong anti-*Campylobacter* effects, whereby the growth temperature and medium, and the presence of oxygen have vital roles in the *B. subtilis* and *C. jejuni* interactions and growth dynamics. In an environment with actively growing *B. subtilis* and *C. jejuni*, such as the chicken gut, we can expect *B. subtilis* to reduce *C. jejuni* growth. However, further in vitro experiments, including a variety of *C. jejuni* strains also resistant to antimicrobials (as would be found on poultry farms), and in vivo experiments including poultry animals are needed to confirm this hypothesis. In contrast, *B. subtilis* might not be suitable for *C. jejuni* control in environments that are unfavorable for *C. jejuni*, as it fails to antagonize the nongrowing pathogen, and might even provide protection for *C. jejuni.*

## 6. Patents

Mandic Mulec I, Simunovic K, Stefanic P, Erega A, Smole Možina S, Klančnik A, Zhang Q, Sahin O. 23 November 2020, filing date. *Bacillus subtilis* strain with strong inhibition of enteropathogenic and foodborne pathogenic bacteria. US patent application 63117215. EFS ID: 41197988, confirmation number: 4289.

## Figures and Tables

**Figure 1 microorganisms-10-00289-f001:**
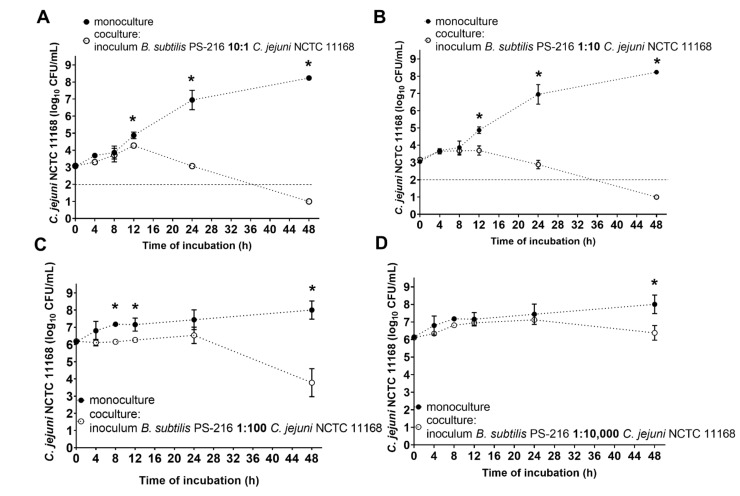
Effects of *B. subtilis* PS-216 coculture on growth of *C. jejuni* NCTC 11168 compared to *C. jejuni* in monoculture, at the starting *B. subtilis*:*C. jejuni* inoculum ratios of 10:1 (**A**), 1:10 (**B**), 1:100 (**C**), and 1:10,000 (**D**) in Mueller-Hinton broth at 42 °C under microaerobic conditions over 48 h. Data are means ± standard deviation of three replicates. Data below the limit of detection (**A**, **B**, dotted line) are given as 1 log_10_CFU/mL. *, *p* < 0.05 (Mann-Whitney U-tests).

**Figure 2 microorganisms-10-00289-f002:**
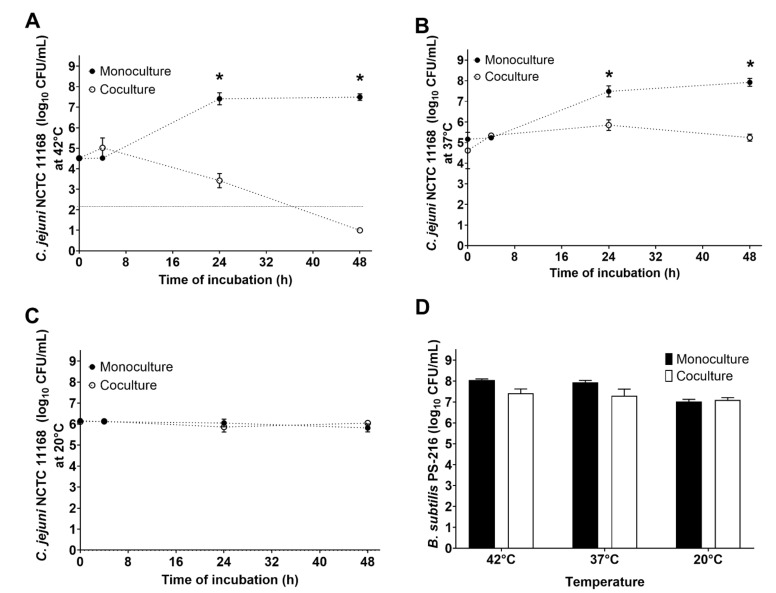
(**A**–**C**) Effects of *B. subtilis* PS-216 coculture on growth of *C. jejuni* NCTC 11168 compared to *C. jejuni* in monoculture, at the starting *B. subtilis*: *C. jejuni* inoculum ratio of 1:10 in Mueller-Hinton broth at 42 °C (**A**), 37 °C (**B**), and 20 °C (**C**) under microaerobic conditions over 48 h. Data below the limit of detection (**A**, dotted line) are given as 1 log_10_CFU/mL. (**D**) *B. subtilis* PS-216 counts under the same conditions as for (**A**–**C**). Data are means ± standard deviation of three replicates. *, *p* < 0.05 (Mann-Whitney U-tests).

**Figure 3 microorganisms-10-00289-f003:**
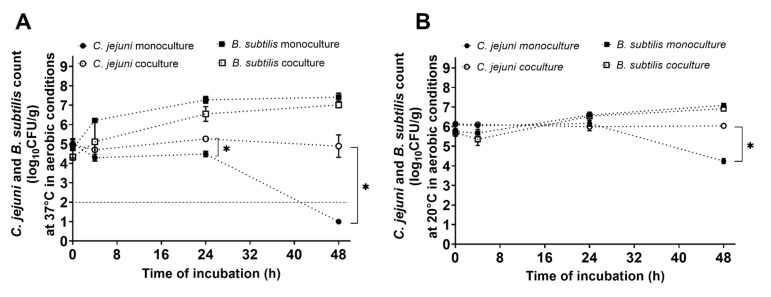
Effects of coculturing on growth of *B. subtilis* PS-216 and *C. jejuni* NCTC 11168 at the starting *B. subtilis*:*C. jejuni* inoculum ratio of 1:10 in Mueller-Hinton broth at 37 °C (**A**) and 20 °C (**B**) under aerobic conditions over 48 h. Data are means ± standard deviation of three replicates. Data below the limit of detection (**A**, dotted line) are given as 1 log_10_CFU/mL. *, *p* < 0.05 (Mann-Whitney U-tests).

**Figure 4 microorganisms-10-00289-f004:**
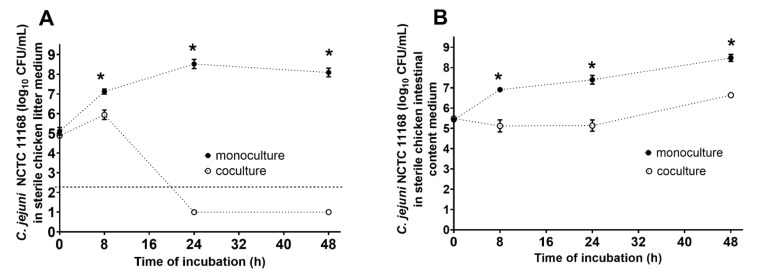
Effects of *B. subtilis* PS-216 coculture on growth of *C. jejuni* NCTC 11168 compared to *C. jejuni* in monoculture at the starting *B. subtilis*: *C. jejuni* inoculum ratio of 1:10 in sterile chicken litter medium (**A**) and sterile chicken intestinal-content medium (**B**) over 48 h. Data are means ± standard deviation of three replicates. Data below the limit of detection (**A**, dotted line) are given as 1 log_10_CFU/mL. *, *p* < 0.05 (Mann–Whitney U-tests).

**Figure 5 microorganisms-10-00289-f005:**
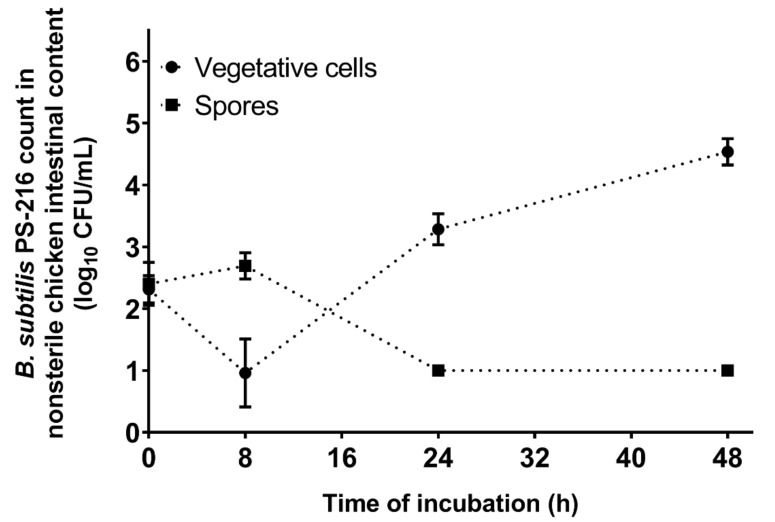
*B. subtilis* PS-216 growth (vegetative cells) and sporulation (spores) in nonsterile chicken intestinal-content medium at 42 °C under microaerobic conditions over 48 h. Data are means ± standard deviation of three replicates.

## Data Availability

Data supporting the reported results can be retrieved from the authors.

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
