# Peer review of "Bacillus subtilis* PS-216 Antagonistic Activities against *Campylobacter jejuni* NCTC 11168 Are Modulated by Temperature, Oxygen, and Growth Medium"

_microorganisms, 2022, doi:10.3390/microorganisms10020289_

Round 1
Reviewer 1 Report
The manuscript entitled “Bacillus subtilis PS-216 antagonistic activities against Campylobacter jejuni 11168 are modulated by temperature, oxygen, and growth medium” by Katarina Šimunović, Polonca Štefanič, Anja Klančnik1, Andi Erega, Ines Mandić Mulec and Sonja Smole Možina, study the relation between a probiotic strain of Bacillus subtilis with Campylobacter jejuni. The manuscript is well written and easy to follow, however there are some aspects of the results/discussion sections that need attention.
Information regarding chicken rearing system (intensive, semi-intensive, extensive) and feed characterisation were not indicated. In my opinion data regarding this information is crucial.
In the discussion section, when relevant, to a better understanding of the data a reference to the adequate figure should be made.
Indicate Campylobacter strain with the full reference NCTC 11168
Lines 128-131: Please, be more detailed in the description of spores enumerations
Line 179: What to the authors mean by “…by disturbed sampling.”
Line 181- Section 2.6
The initial concentration of B. subtilis PS-216 is not indicated. Also, there are no indication if the inoculum has only vegetative cells or both spores and vegetative cells.
Line 230: One should not expect for any sort of data, so the expression “As expected…” should be changed.
Line 303: Figure 5- Please explain why the counts of spores and vegetative cells of B. subtilis are the same at time 0h. See comment related to line 181.
Line 328-329: This sentence should be modified. Since Campylobacter jejuni is not growing due to unfavourable culture conditions the effect of B. subtilis can not be assessed.
Lines 347-249: Consider rephrasing. Further explain why is that “…the protective effects seen here for B. subtilis under aerobic conditions are not surprising”?
Lines 365- 366: Where are the data that support this affirmation? The only graph I can find with regards to the growth on chicken litter regards Campylobacter and not Bacillus.
Line 372: Where are the data referring to the control growth of B. subtilis in chicken intestine content medium?
Figure 2, D- Why were the B. subtilis counts only made in the temperatures assay?
Author Response
The authors would like to thank you for the thorough review of the manuscript and the constructive feedback that helped us to improve the quality of our work. All points are addressed below and appropriate changes have been made to the manuscript using the 'track changes' option in MS Word. Responses to comments are provided below. All line numbers given in the answers refer to the lines when the manuscript is shown as 'all markup'. We hope that the changes made to the manuscript are satisfactory.
Reviewer 1:
The manuscript entitled “Bacillus subtilis PS-216 antagonistic activities against Campylobacter jejuni 11168 are modulated by temperature, oxygen, and growth medium” by Katarina Šimunović, Polonca Štefanič, Anja Klančnik1, Andi Erega, Ines Mandić Mulec and Sonja Smole Možina, study the relation between a probiotic strain of Bacillus subtilis with Campylobacter jejuni. The manuscript is well written and easy to follow, however there are some aspects of the results/discussion sections that need attention.
Comment:
Information regarding chicken rearing system (intensive, semi-intensive, extensive) and feed characterisation were not indicated. In my opinion data regarding this information is crucial.
Response:
Additional information on the rearing of these broilers was added L184-186 for clarification. The broilers were 49 days old at the time of slaughter, came from a semi-intensive rearing system, and were fed standard feed consisting of starter and finisher diets bought from a commercial feed supplier, Jata Emona d.o.o.
Comment:
In the discussion section, when relevant, to a better understanding of the data a reference to the adequate figure should be made.
Response:
References to figures were added in L346 (Figure 2), L361 (Figure 3), L389 (Figure S2A), L396 (Figure 4), L399 (Figure S2B) and L426-427 (Figure 5)
Comment:
Indicate Campylobacter strain with the full reference NCTC 11168
The full reference NCTC was added to C. jejuni 11168 designation in lines: L3 (title), L16 and L27 (abstract), L143, L156, L167, L239, L263, L269, L280, L307, L335, and L452.
Comment:
Lines 128-131: Please, be more detailed in the description of spores enumerations
Response:
The spore enumeration procedure was described in more detail in L135-140.
Comment:
Line 179: What to the authors mean by “…by disturbed sampling.”
Response:
The sampling procedure was described in more detail in L192-195, to clarify the term ‘disturbed sampling’.
Comment:
Line 181- Section 2.6
The initial concentration of B. subtilis PS-216 is not indicated. Also, there are no indication if the inoculum has only vegetative cells or both spores and vegetative cells.
Response:
We have described this part for the procedure in more detail and added the initial number of both spores and vegetative cells in L201-203.
Comment:
Line 230: One should not expect for any sort of data, so the expression “As expected…” should be changed.
Response:
We thank you for this remark. You are absolutely right and the text has been removed (L248).
Comment:
Line 303: Figure 5- Please explain why the counts of spores and vegetative cells of B. subtilis are the same at time 0h. See comment related to line 181.
Response:
The counts are not really identical, but very similar. The total inoculum consists of approximately 3x102 spores and 2x102 vegetative cells. Additional information was added in L201-203.
Comment:
Line 328-329: This sentence should be modified. Since Campylobacter jejuni is not growing due to unfavourable culture conditions the effect of B. subtilis can not be assessed.
Response:
The sentence has been revised somewhat to clarify our statement in L348-349. We realize that it cannot be said to inhibit growth (since C. jejuni did not grow), but no antagonistic effect on C. jejuni was observed (the concentration remained the same).
Comment:
Lines 347-249: Consider rephrasing. Further explain why is that “…the protective effects seen here for B. subtilis under aerobic conditions are not surprising”?
Response:
The sentence has been modified in L369-L372, to clarify our claim: "Under these conditions where the oxygen levels would be toxic for C. jejuni, there would not be any apparent antagonistic effects, although B. subtilis would still reduce the oxygen in the environment. This might therefore be an issue for food safety, as it might promote C. jejuni survival and persistence in the environment".
Comment:
Lines 365- 366: Where are the data that support this affirmation? The only graph I can find with regards to the growth on chicken litter regards Campylobacter and not Bacillus.
Response:
This information, as most B. subtilis growth results, is provided as supplement, in Figure S2A. We have added a reference to the figure in L389.
Comment:
Line 372: Where are the data referring to the control growth of B. subtilis in chicken intestine content medium?
Response:
Most growth results for B. subtilis was provided as supplemental information as the co-cultivation did not affect B. subtilis growth. We have added a reference to Figure S2B in L399.
Comment:
Figure 2, D- Why were the B. subtilis counts only made in the temperatures assay?
Response:
In Figure 2, we show that the growth of B. subtilis in co-cultures with C. jejuni is somewhat but not significantly affected. All other growth data of B. subtilis can be found as supplemental information. These data can be found in Figure S1.
Reviewer 2 Report
In this manuscript, Šimunović and colleagues investigated how B. subtilis PS-216 reduce C. jejuni 11168 growth in vitro. The findings are interesting, but several questions should be addressed. The following are the specific concerns.
- 11168 is able to colonize chicken gut and enumerated in chicken digesta or fecal. The result of 11168 unable to grow in non-sterile intestinal content in mono-culture (lines 300-302) should be presented, possibly at supplemental. This observation raises the possibility of mutant/weaken C jejuni 11168 isolate in this study. It is necessary to use a new 11168 isolate or a new strain, such as 81176 or some chicken strains, to be able to grow in the non-sterile intestinal content. It is necessary to repeat some of the key findings using the new isolate(s).
- Different initial C. jejuni doses were used in different figures. What is the reasoning to do this way? Was it possible that C. jejuni at certain CFU/ml (e.g. 10^6-7 might resist to B. subtilis? In normal chicken gut, C. jejuni can be counted as high as 10^8 CFU/ml(g).
- At lines 23-24, 26-27, please revise the sentences because the results didn’t support the claims. In fact, chicken content was better than B. subtilis to reduce C. jejuni, based on your not shown results.
- At lines 135-144, please detail how you estimated the CFU of B. subtilis and C. jejuni. Did you use formula of OD reading and CFU?
- At lines 172-173, what age of the broilers are used? Did the broilers feed/water have antibiotics or any drugs?
Reviewer 3 Report
The manuscript is an important contribution to the efforts to find non-antimocrobial control/mitigation strategies to reduce the intestinal Campylobacter load of broiler chicken, which, in turn, can contribute to a high degree to the reduction of Campylobacteriosis in humans. The results of the own studies offer a very good basis for further investigations into the potential of special strains of B. subtilis to compete with Campylobacter jejuni in the gut of broilers (and possibly even in poultry litter).
Two minor comments:
- It may be favourable to ask a native speaker for a final proof-read: the English langauge is for authors as non native speakers close to excellent. However, there are some expressions that highly probably would be somewhat different, when written by a native speaker. Only as examples: Line 84: the expression "...key ecological factor.." sounds as if "...ecological key factor" is linguistically correct. And in Line 300: "... was examined in these conditions.." should read "...was examnined under these conditions..".
- Line 311: to call the attempts to suppress Campylobacter jejuni by a competitive growth of B. subtilis "...a strategy of competitive exclusion" is not correct. The "competitive exclusion strategy is based on implementing a probiotic mixture into teh naive gut of freshly hatched chickes before the natural microbiome is building up in the chickens' guts, which makes the growth of Salmonella strains difficult (which "excludes Salomella". The mode of action of B. subtilis is not "exclusion" but suppressing Campylobacter.
However, all in all, it is a very valuable manuscript.
Author Response
Authors would like to express their appreciation for the thorough review of the manuscript and the positive and constructive feedback that helped us to improve the quality of our paper. In the following, all points are addressed and appropriate changes have been made in the manuscript with the ‘track changes’ option in MS Word. Below are the responses to the comments. All indicated line numbers in the answers are related to the lines when the manuscript is viewed as ‘all markup’. We hope the changes made to the manuscript are satisfactory.
Reviewer 3:
The manuscript is an important contribution to the efforts to find non-antimicrobial control/mitigation strategies to reduce the intestinal Campylobacter load of broiler chicken, which, in turn, can contribute to a high degree to the reduction of Campylobacteriosis in humans. The results of the own studies offer a very good basis for further investigations into the potential of special strains of B. subtilis to compete with Campylobacter jejuni in the gut of broilers (and possibly even in poultry litter).
Two minor comments:
Comment:
- It may be favourable to ask a native speaker for a final proof-read: the English langauge is for authors as non native speakers close to excellent. However, there are some expressions that highly probably would be somewhat different, when written by a native speaker. Only as examples: Line 84: the expression "...key ecological factor.." sounds as if "...ecological key factor" is linguistically correct. And in Line 300: "... was examined in these conditions.." should read "...was examnined under these conditions.."
Response:
A native speaker has checked and proofread the manuscript.
Comment:
- Line 311: to call the attempts to suppress Campylobacter jejuni by a competitive growth of B. subtilis "...a strategy of competitive exclusion" is not correct. The "competitive exclusion strategy is based on implementing a probiotic mixture into teh naive gut of freshly hatched chickes before the natural microbiome is building up in the chickens' guts, which makes the growth of Salmonella strains difficult (which "excludes Salomella". The mode of action of B. subtilis is not "exclusion" but suppressing Campylobacter.
Response:
Yes, we agree with the suggestion and the sentence has been changed accordingly in L330.
However, all in all, it is a very valuable manuscript.
Round 2
Reviewer 2 Report
The authors have addressed most of my concerns, hence, I suggest accepting the manuscript in the current form.Author Response
The authors would like to thank the reviewer for their invested time, the constructive review and a positive and timely response to the changes made.